# AUXILIARY GUIDED AUTOREGRESSIVE VARIATIONAL AUTOENCODERS

## ABSTRACT

Generative modeling of high-dimensional data is a key problem in machine learning. Successful approaches include latent variable models and autoregressive models. The complementary strengths of these approaches, to model global and local image statistics respectively, suggest hybrid models combining the strengths of both models. Our contribution is to train such hybrid models using an auxiliary loss function that controls which information is captured by the latent variables and what is left to the autoregressive decoder. In contrast, prior work on such hybrid models needed to limit the capacity of the autoregressive decoder to prevent degenerate models that ignore the latent variables and only rely on autoregressive modeling. Our approach results in models with meaningful latent variable representations, and which rely on powerful autoregressive decoders to model image details. Our model generates qualitatively convincing samples, and yields state-of-the-art quantitative results.

## 1 INTRODUCTION

Unsupervised modeling of complex distributions with unknown structure is a landmark challenge in machine learning. The problem is often studied in the context of learning generative models of the complex high-dimensional distributions of natural image collections. Latent variable approaches can learn disentangled and concise representations of the data (Bengio et al., 2013), which are useful for compression (Gregor et al., 2016) and semi-supervised learning (Kingma et al., 2014; Rasmus et al., 2015). When conditioned on prior information, generative models can be used for a variety of tasks, such as attribute or class-conditional image generation, text and pose-based image generation, image colorization, *etc.* (Yan et al., 2016; van den Oord et al., 2016; Reed et al., 2017; Deshpande et al., 2017). Recently significant advances in generative (image) modeling have been made along several lines, including adversarial networks (Goodfellow et al., 2014; Arjovsky et al., 2017), variational autoencoders (Kingma & Welling, 2014; Rezende et al., 2014), autoregressive models (Oord et al., 2016; Reed et al., 2017), and non-volume preserving variable transformations (Dinh et al., 2017).

In our work we seek to combine the merits of two of these lines of work. Variational autoencoders (VAEs) (Kingma & Welling, 2014; Rezende et al., 2014) can learn latent variable representations that abstract away from low-level details, but model pixels as conditionally independent given the latent variables. This renders the generative model computationally efficient, but the lack of low-level structure modeling leads to overly smooth and blurry samples. Autoregressive models, such as pixelCNNs (Oord et al., 2016), on the other hand, estimate complex translation invariant conditional distributions among pixels. They are effective to model low-level image statistics, and yield state-of-the-art likelihoods on test data (Salimans et al., 2017). This is in line with the observations of Kolesnikov & Lampert (2017) that low-level image details account for a large part of the likelihood. These autoregressive models, however, do not learn a latent variable representations to support, *e.g.*, semi-supervised learning. See Figure 1 for representative samples of VAE and pixelCNN models.

The complementary strengths of VAEs and pixelCNNs, modeling global and local image statistics respectively, suggest hybrid approaches combining the strengths of both. Prior work on such hybrid models needed to limit the capacity of the autoregressive decoder to prevent degenerate models that completely ignore the latent variables and rely on autoregressive modeling only (Gulrajani et al., 2017; Chen et al., 2017). In this paper we describe Auxiliary Guided Autoregressive Variational autoEncoders (AGAVE), an approach to train such hybrid models using an auxiliary loss function

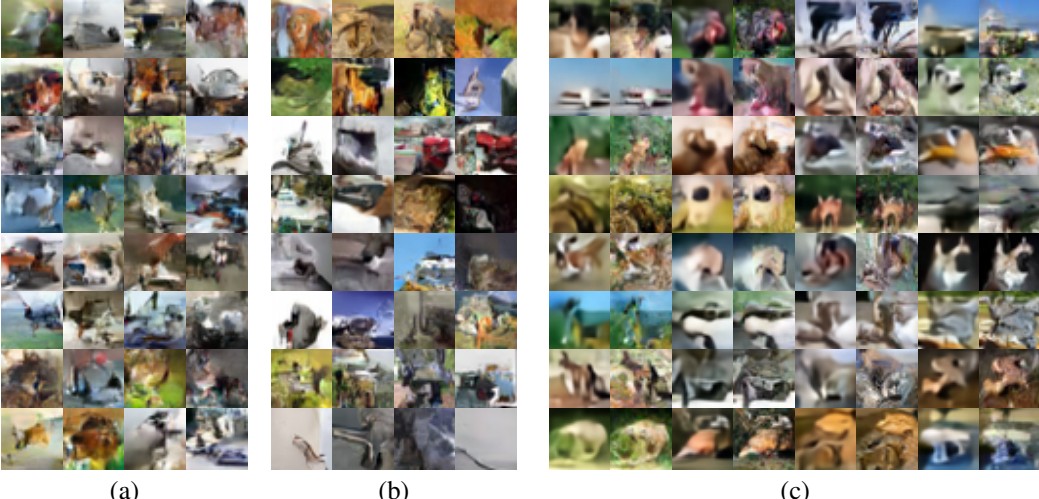

(a)                (b)                          (c)

Figure 1: Randomly selected samples from unsupervised models trained on $32\times32$ CIFAR10 images: (a) IAF-VAE Kingma et al. (2016), (b) pixelCNN++ Salimans et al. (2017), and (c) our hybrid AGAVE model. For our model, we show the intermediate high-level representation based on latent variables (left), that conditions the final sample based on the pixelCNN decoder (right).

that controls which information is captured by the latent variables and what is left to the AR decoder, rather than limiting the capacity of the latter. Using high-capacity VAE and autoregressive components allows our models to obtain quantitative results on held-out data that are on par with the state of the art, and to generate samples with both global coherence and low-level details, see Figure 1.

## 2 RELATED WORK

Generative image modeling has recently taken significant strides forward, leveraging deep neural networks to learn complex density models using a variety of approaches. These include the variational autoencoders and autoregressive models that form the basis of our work, but also generative adversarial networks (GANs) (Goodfellow et al., 2014; Arjovsky et al., 2017) and variable transformation with invertible functions (Dinh et al., 2017). While GANs produce visually appealing samples, they suffer from mode dropping and their likelihood-free nature prevents measuring how well they model held-out test data. In particular, GANs can only generate samples on a non-linear manifold in the data space with dimension equal to the number of latent variables. In contrast, probabilistic models such as VAEs and autoregressive models generalize to the entire data space, and likelihoods of held-out data can be used for compression, and to quantitatively compare different models. The non-volume preserving (NVP) transformation approach of Dinh et al. (2017) chains together invertible transformations to map a basic (*e.g.* unit Gaussian) prior on the latent space to a complex distribution on the data space. This method offers tractable likelihood evaluation and exact inference, but obtains likelihoods on held-out data below the values reported using state-of-the-art VAE and autoregressive models. Moreover, it is restricted to use latent representations with the same dimensionality as the input data, and is thus difficult to scale to model high-resolution images.

Autoregressive density estimation models, such as pixelCNNs (Oord et al., 2016), admit tractable likelihood evaluation, while for variational autoencoders (Kingma & Welling, 2014; Rezende et al., 2014) accurate approximations can be obtained using importance sampling (Burda et al., 2016). Naively combining powerful pixelCNN decoders in a VAE framework results in a degenerate model which ignores the VAE latent variable structure, as explained through the lens of bits-back coding by Chen et al. (2017). To address this issue, the capacity of the the autoregressive component can be restricted. This can, for example, be achieved by reducing its depth and/or field of view, or by giving the pixelCNN only access to grayscale values, *i.e.* modeling $p(x_i|\mathbf{x}_{<i}, \mathbf{z}) = p(x_i|\text{gray}(\mathbf{x}_{<i}), \mathbf{z})$ (Chen et al., 2017; Gulrajani et al., 2017). This forces the model to leverage the latent variables $\mathbf{z}$ to model part of the dependencies among the pixels. This approach, however, has two drawbacks.

(i) Curbing the capacity of the model is undesirable in unsupervised settings where training data is abundant and overfitting unlikely. (ii) Balancing what is modeled by the VAE and the pixelCNN by means of architectural design choices requires careful hand-design and tuning of the architectures. To overcome these drawbacks, we propose to instead control what is modeled by the VAE and pixelCNN with an auxiliary loss on the VAE decoder output before it is used to condition the autoregressive decoder. This allows us to "plug in" powerful high-capacity VAE and pixelCNN architectures, and balance what is modeled by each component by means of the auxiliary loss.

In a similar vein, Kolesnikov & Lampert (2017) force pixelCNN models to capture more high-level image aspects using an auxiliary representation $\mathbf{y}$ of the original image $\mathbf{x}$, *e.g.* a low-resolution version of the original. They learn a pixelCNN for $\mathbf{y}$, and a conditional pixelCNN to predict $\mathbf{x}$ from $\mathbf{y}$, possibly using several intermediate representations. This approach forces modeling of more high-level aspects in the intermediate representations, and yields visually more compelling samples. Reed et al. (2017) similarly learn a series of conditional autoregressive models to upsample coarser intermediate latent images. By introducing partial conditional independencies in the model they scale the model to efficiently sample high-resolution images of up to $512{\times}512$ pixels. Gregor et al. (2016) use a recurrent VAE model to produces a sequence of RGB images with increasing detail derived from latent variables associated with each iteration. Like our work, all these models work with intermediate representations in RGB space to learn accurate generative image models.

## 3 Auxiliary guided autoregressive variational autoencoders

We give a brief overview of variational autoencoders and their limitations in Section 3.1, before we present our approach to learn variational autoencoders with autoregressive decoders in Section 3.2.

### 3.1 Variational autoencoders

Variational autoencoders (Kingma & Welling, 2014; Rezende et al., 2014) learn deep generative latent variable models using two neural networks. The "decoder" network implements a conditional distribution $p_{\boldsymbol{\theta}}(\mathbf{x}|\mathbf{z})$ over observations $\mathbf{x}$ given a latent variable $\mathbf{z}$, with parameters $\boldsymbol{\theta}$. Together with a basic prior on the latent variable $\mathbf{z}$, *e.g.* a unit Gaussian, the generative model on $\mathbf{x}$ is obtained by marginalizing out the latent variable:

$$p_{\boldsymbol{\theta}}(\mathbf{x}) = \int p(\mathbf{z})p_{\boldsymbol{\theta}}(\mathbf{x}|\mathbf{z}) \, d\mathbf{z}. \tag{1}$$

The marginal likelihood can, however, not be optimized directly since the non-linear dependencies in $p_{\boldsymbol{\theta}}(\mathbf{x}|\mathbf{z})$ render the integral intractable. To overcome this problem, an "encoder" network is used to compute an approximate posterior distribution $q_{\boldsymbol{\phi}}(\mathbf{z}|\mathbf{x})$, with parameters $\boldsymbol{\phi}$. The approximate posterior is used to define a variational bound on the data log-likelihood, by subtracting the Kullback-Leibler divergence between the true and approximate posterior:

$$\ln p_{\boldsymbol{\theta}}(\mathbf{x}) \geq \mathcal{L}(\boldsymbol{\theta}, \boldsymbol{\phi}; \mathbf{x}) \quad = \quad \ln(p_{\boldsymbol{\theta}}(\mathbf{x})) - D_{\text{KL}}(q_{\boldsymbol{\phi}}(\mathbf{z}|\mathbf{x})||p_{\boldsymbol{\theta}}(\mathbf{z}|\mathbf{x})) \tag{2}$$

$$= \quad \underbrace{\mathbb{E}_{q_{\boldsymbol{\phi}}}[\ln(p_{\boldsymbol{\theta}}(\mathbf{x}|\mathbf{z})]}_{\text{Reconstruction}} - \underbrace{D_{\text{KL}}(q_{\boldsymbol{\phi}}(\mathbf{z}|\mathbf{x})||p(\mathbf{z}))}_{\text{Regularization}}. \tag{3}$$

The decomposition in (3) interprets the bound as the sum of a reconstruction term and a regularization term. The first aims to maximize the expected data log-likelihood $p_{\boldsymbol{\theta}}(\mathbf{x}|\mathbf{z})$ given the posterior estimate $q_{\boldsymbol{\phi}}(\mathbf{z}|\mathbf{x})$. The second term prevents $q_{\boldsymbol{\phi}}(\mathbf{z}|\mathbf{x})$ from collapsing to a single point, which would be optimal for the first term.

Variational autoencoders typically model the dimensions of $\mathbf{x}$ as conditionally independent,

$$p_{\boldsymbol{\theta}}(\mathbf{x}|\mathbf{z}) = \prod_{i=1}^{D} p_{\boldsymbol{\theta}}(x_i|\mathbf{z}), \tag{4}$$

for instance using a factored Gaussian or Bernoulli model, see *e.g.* Kingma & Welling (2014); Kingma et al. (2016); Yan et al. (2016). The conditional independence assumption makes sampling from the VAE efficient: since the decoder network is evaluated only once for a sample $\mathbf{z} \sim p(\mathbf{z})$ to compute all the conditional distributions $p_{\boldsymbol{\theta}}(x_i|\mathbf{z})$, the $x_i$ can then be sampled in parallel.

A result of relying on the latent variables to account for all pixel dependencies, however, is that all low-level variability must also be modeled by the latent variables. Consider, for instance, a picture of a dog, and variants of that image shifted by one or a few pixels, or in a slightly different pose, with a slightly lighter background, or with less saturated colors, *etc.* If these factors of variability are modeled using latent variables, then these low-level aspects are confounded with latent variables relating to the high-level image content. If the corresponding image variability is not modeled using latent variables, it will be modeled as independent pixel noise. In the latter case, using the mean of $p_{\boldsymbol{\theta}}(\mathbf{x}|\mathbf{z})$ as the synthetic image for a given $\mathbf{z}$ results in blurry samples, since the mean is averaged over the low-level variants of the image. Sampling from $p_{\boldsymbol{\theta}}(\mathbf{x}|\mathbf{z})$ to obtain synthetic images, on the other hand, results in images with unrealistic independent pixel noise.

## 3.2 AUTOREGRESSIVE DECODERS IN VARIATIONAL AUTOENCODERS

Autoregressive density models, see *e.g.* (Larochelle & Murray, 2011; Germain et al., 2015), rely on the basic factorization of multi-variate distributions,

$$p_{\boldsymbol{\theta}}(\mathbf{x}) = \prod_{i=1}^{D} p_{\boldsymbol{\theta}}(x_i|\mathbf{x}_{<i}) \tag{5}$$

with $\mathbf{x}_{<i} = x_1, \ldots, x_{i-1}$, and model the conditional distributions using a (deep) neural network. For image data, PixelCNNs (Oord et al., 2016; van den Oord et al., 2016) use a scanline pixel ordering, and model the conditional distributions using a convolution neural network. The convolutional filters are masked so as to ensure that the receptive fields only extend to pixels $\mathbf{x}_{<i}$ when computing the conditional distribution of $x_i$.

PixelCNNs can be used as a decoder in a VAE by conditioning on the latent variable $\mathbf{z}$ in addition to the preceding pixels, leading to a variational bound with a modified reconstruction term:

$$\mathcal{L}(\boldsymbol{\theta}, \boldsymbol{\phi}; \mathbf{x}) = \mathbb{E}_{q_{\boldsymbol{\phi}}} \left[ \sum_{i=1}^{D} \ln p_{\boldsymbol{\theta}}(x_i|\mathbf{x}_{<i}, \mathbf{z}) \right] - D_{\mathrm{KL}}(q_{\boldsymbol{\phi}}(\mathbf{z}|\mathbf{x})||p(\mathbf{z})). \tag{6}$$

The regularization term can be interpreted as a "cost" of using the latent variables. To effectively use the latent variables, the approximate posterior $q_{\boldsymbol{\phi}}(\mathbf{z}|\mathbf{x})$ must differ from the prior $p(\mathbf{z})$, which increases the KL divergence.

Chen et al. (2017) showed that for loss in (6) and a decoder with enough capacity, it is optimal to encode no information about $x$ in $z$ by setting $q(z|x) = p(z)$. To ensure meaningful latent representation learning Chen et al. (2017) and Gulrajani et al. (2017) restrict the capacity of the pixelCNN decoder. In our approach, in contrast, it is always optimal for the autoregressive decoder, regardless of its capacity, to exploit the information on $\mathbf{x}$ carried by $z$. We rely on two decoders in parallel: the first one reconstructs an auxiliary image $\mathbf{y}$ from an intermediate representation $f_{\boldsymbol{\theta}}(\mathbf{z})$ in a non-autoregressive manner. The auxiliary image can be either simply taken to be the original image ($\mathbf{y} = \mathbf{x}$), or a compressed version of it, *e.g.* with lower resolution or with a coarser color quantization. The second decoder is a conditional autoregressive model that predicts $\mathbf{x}$ conditioned on $f_{\boldsymbol{\theta}}(\mathbf{z})$. Modeling $\mathbf{y}$ in a non-autoregressive manner ensures a meaningful representation $\mathbf{z}$ and renders $\mathbf{x}$ and $\mathbf{z}$ dependent, inducing a certain non-zero KL "cost" in (6). The uncertainty on $\mathbf{x}$ is thus reduced when conditioning on $\mathbf{z}$, and there is no longer an advantage in ignoring the latent variable for the autoregressive decoder. We provide a more detailed explanation of why our auxiliary loss ensures a meaningful use of latent variables in powerful decoders in Appendix A. To train the model we combine both decoders in a single objective function with a shared encoder network:

$$\mathcal{L}(\boldsymbol{\theta}, \boldsymbol{\phi}; \mathbf{x}, \mathbf{y}) = \underbrace{\mathbb{E}_{q_{\boldsymbol{\phi}}} \left[ \sum_{i=1}^{D} \ln p_{\boldsymbol{\theta}}(x_i|\mathbf{x}_{<i}, \mathbf{z}) \right]}_{\text{Primary Reconstruction}} + \underbrace{\mathbb{E}_{q_{\boldsymbol{\phi}}} \left[ \sum_{j=1}^{E} \ln p_{\boldsymbol{\theta}}(y_j|\mathbf{z}) \right]}_{\text{Auxiliary Reconstruction}} - \underbrace{\lambda \, D_{\mathrm{KL}} \left( q_{\boldsymbol{\phi}}(\mathbf{z}|\mathbf{x})||p(\mathbf{z}) \right)}_{\text{Regularization}}. \tag{7}$$

Treating $\mathbf{x}$ and $\mathbf{y}$ as two variables that are conditionally independent given a shared underlying latent vairable $\mathbf{z}$ leads to $\lambda = 1$. Summing the lower bounds in Eq. (3) and Eq. (6) of the marginal log-likelihoods of $\mathbf{y}$ and $\mathbf{x}$, and sharing the encoder network, leads to $\lambda = 2$. Larger values of $\lambda$ result in valid but less tight lower bounds of the log-likelihoods. Encouraging the variational posterior to be closer to the prior, this leads to less informative latent variable representations.

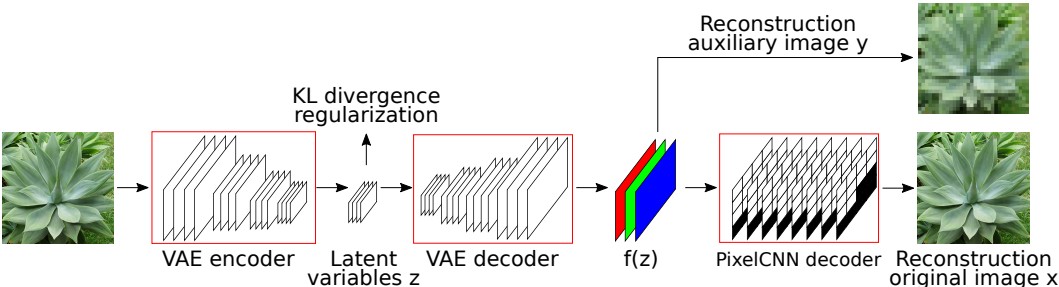

Figure 2: Schematic illustration of our auxiliary guided autoregressive variational autoencoder (AGAVE). The objective function has three components: KL divergence regularization, per-pixel reconstruction with the VAE decoder, and autoregressive reconstruction with the pixelCNN decoder.

Sharing the encoder across the two decoders is the key of our approach. The factored auxiliary VAE decoder can only model pixel dependencies by means of the latent variables, which ensures that a meaningful representation is learned. Now, given that the VAE encoder output is informative on the image content, there is no incentive for the autoregressive decoder to ignore the intermediate representation $f(\mathbf{z})$ on which it is conditioned. The choice of the regularization parameter $\lambda$ and auxiliary image $\mathbf{y}$ provide two levers to control *how much* and *what type* of information should be encoded in the latent variables. See Figure 2 for a schematic illustration of our approach.

## 4 EXPERIMENTAL EVALUATION

In this section we describe our experimental setup, and present results on CIFAR10.

### 4.1 DATASET AND IMPLEMENTATION

The CIFAR10 dataset (Krizhevsky, 2009) contains 6,000 images of $32{\times}32$ pixels for each of the 10 object categories *airplane, automobile, bird, cat, deer, dog, frog, horse, ship, truck*. The images are split into 50,000 training images and 10,000 test images. We train all our models in a completely unsupervised manner, ignoring the class information.

We implemented our model based on existing architectures. In particular we use the VAE architecture of Kingma et al. (2016), and use logistic distributions over the RGB color values. We let the intermediate representation $f(\mathbf{z})$ output by the VAE decoder be the per-pixel and per-channel mean values of the logistics, and learn per-channel scale parameters that are used across all pixels. The cumulative density function (CDF), given by the sigmoid function, is used to compute probabilities across the 256 discrete color levels, or fewer if a lower quantization level is chosen in $\mathbf{y}$. Using RGB values $y_i \in [0, 255]$, we let $b$ denote the number of discrete color levels and define $c = 256/b$. The probabilities over the $b$ discrete color levels are computed from the logistic mean and variance $\mu_i$ and $s_i$ as

$$p(y_i|\mu_i, s_i) = \sigma\left(c + c\lfloor y_i/c\rfloor|\mu_i, s_i\right) - \sigma\left(c\lfloor y_i/c\rfloor|\mu_i, s_i\right). \tag{8}$$

For the pixelCNN we use the architecture of Salimans et al. (2017), and modify it to be conditioned on the VAE decoder output $f(\mathbf{z})$, or possibly an upsampled version if $\mathbf{y}$ has a lower resolution than $\mathbf{x}$. In particular, we apply standard non-masked convolutional layers to the VAE output, as many as there are pixelCNN layers. We allow each layer of the pixel-CNN to take additional input using non-masked convolutions from the feature stream based on the VAE output. This ensures that the conditional pixelCNN remains autoregressive.

To speed up training, we independently pretrain the VAE and pixelCNN in parallel, and then continue training the full model with both decoders. We use the Adamax optimizer (Kingma & Ba, 2015) with a learning rate of 0.002 without learning rate decay. We will release our TensorFlow-based code to replicate our experiments upon publication.

| Model | | BPD | .\|z | .\|x_{j<i} |
|---|---|---|---|---|
| NICE | (Dinh et al., 2015) | 4.48 | ✓ | |
| Conv. DRAW | (Gregor et al., 2016) | ≤ 3.58 | ✓ | |
| Real NVP | (Dinh et al., 2017) | 3.49 | ✓ | |
| MatNet | (Bachman, 2016) | ≤ 3.24 | ✓ | |
| PixelCNN | (Oord et al., 2016) | 3.14 | | ✓ |
| VAE-IAF | (Kingma et al., 2016) | ≤ 3.11 | ✓ | |
| Gated pixelCNN | (van den Oord et al., 2016) | 3.03 | | ✓ |
| Pixel-RNN | (Oord et al., 2016) | 3.00 | | ✓ |
| Aux. pixelCNN | (Kolesnikov & Lampert, 2017) | 2.98 | | ✓ |
| Lossy VAE | (Chen et al., 2017) | ≤ 2.95 | ✓ | ✓ |
| **AGAVE**, $\lambda = 12$ | (this paper) | ≤ 2.92 | ✓ | ✓ |
| pixCNN++ | (Salimans et al., 2017) | 2.92 | | ✓ |

Table 1: Bits per dimension (lower is better) of models on the CIFAR10 test data.

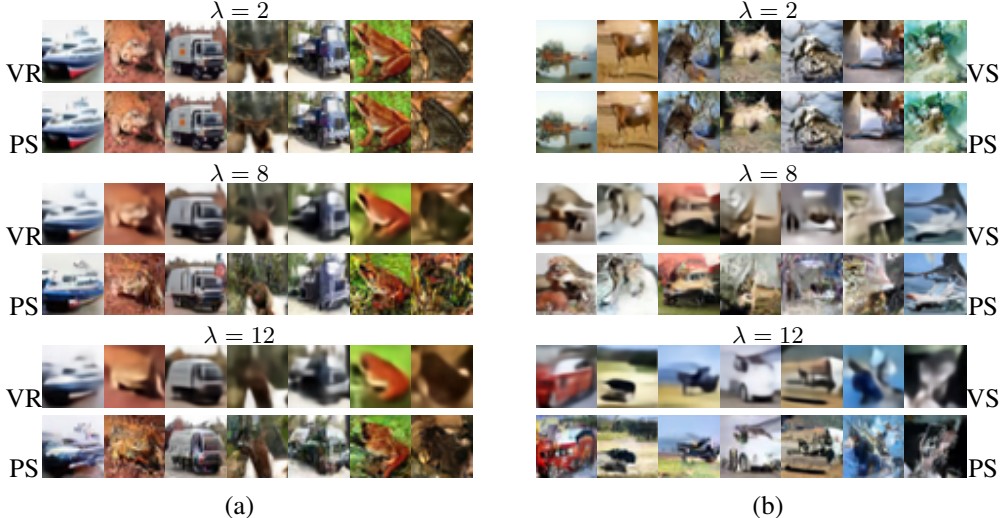

(a)     (b)

Figure 3: Effect of the regularization parameter $\lambda$. Reconstructions (a) and samples (b) of the VAE decoder (VR and VS, respectively) and corresponding conditional samples from the pixelCNN (PS).

## 4.2 EXPERIMENTAL RESULTS

**Quantitative performance evaluation.** Following previous work, we evaluate models on the test images using the bits-per-dimension (BPD) metric: the negative log-likelihood divided by the number of pixels values ($3 \times 32 \times 32$). It can be interpreted as the average number of bits per RGB value in a lossless compression scheme derived from the model.

The comparison in Table 1 shows that our model performs on par with the state-of-the-art results of the pixelCNN++ model (Salimans et al., 2017). Here we used the importance sampling-based bound of Burda et al. (2016) with 150 samples to compute the BPD metric for our model.[1] We refer to Figure 1 for qualitative comparison of samples from our model and pixelCNN++, the latter generated using the publicly available code.

**Effect of KL regularization strength.** In Figure 3 we show reconstructions of test images and samples generated by the VAE decoder, together with their corresponding conditional pixelCNN samples for different values of $\lambda$. As expected, the VAE reconstructions become less accurate for larger values of $\lambda$, mainly by lacking details while preserving the global shape of the input. At the same time, the samples become more appealing for larger $\lambda$, suppressing the unrealistic high-frequency detail in the VAE samples obtained at lower values of $\lambda$. Note that the VAE samples and

---

[1]The graphs in Figure 4 and Figure 5 are based on the bound in Eq. (7) to reduce the computational effort.

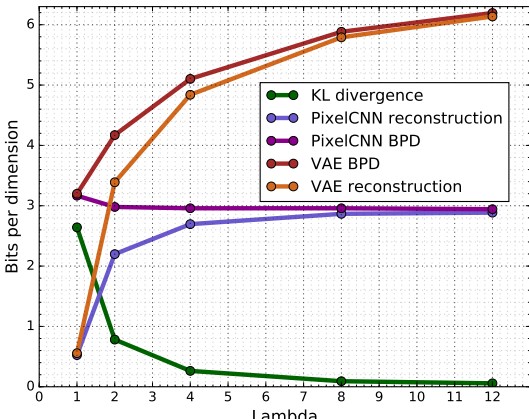

Figure 4: Bits per dimension of the VAE decoder and pixelCNN decoder, as well as decomposition in KL regularization and reconstruction terms.

reconstructions become more similar as $\lambda$ increases, which makes the input to the pixelCNN during training and sampling more consistent.

For both reconstructions and samples, the pixelCNN clearly takes into account the output of the VAE decoder, demonstrating the effectiveness of our auxiliary loss to condition high-capacity pixelCNN decoders on latent variable representations. Samples from the pixelCNN faithfully reproduce the global structure of the VAE output, leading to more realistic samples, in particular for higher values of $\lambda$. In Appendix B we provide more samples of the autoregressive component conditioned on the same output of the VAE decoder. These confirm the dependence on the intermediate representation $f(\mathbf{z})$. Moreover, the samples show that their variability is structured, and could not be modeled by a simple factored decoder.

For $\lambda = 2$ the VAE reconstructions are near perfect during training, and the pixelCNN decoder does not significantly modify the appearance of the VAE output. For larger values of $\lambda$, the pixelCNN clearly adds significant detail to the VAE outputs.

Figure 4 traces the BPD metrics of both the VAE and pixelCNN decoder as a function of $\lambda$. We also show the decomposition in regularization and reconstruction terms. By increasing $\lambda$, the KL divergence can be pushed closer to zero. As the KL divergence term drops, the reconstruction term for the VAE rapidly increases and the VAE model obtains worse BPD values, stemming from the inability of the VAE to model pixel dependencies other than via the latent variables. The reconstruction term of the pixelCNN decoder also increases with $\lambda$, as the amount of information it receives drops. However, in terms of BPD which sums KL divergence and pixelCNN reconstruction, a substantial gain of 0.2 is observed increasing $\lambda$ from 1 to 2, after which smaller but consistent gains are observed.

In Appendix C we present the results of a control experiment where during training we first optimize our objective function in Eq. (7), *i.e.* including the auxiliary reconstruction term, and then switch to optimize the standard objective function of Eq. (6) without the auxiliary term. This leads to models that completely ignore the latent variable structure, as predicted by the analysis of Chen et al. (2017).

**Effect of different auxiliary images.**   We assess the effect of using coarser RGB quantizations and lower spatial resolutions in the auxiliary image. Both make the VAE reconstruction task easier, and transfer modeling of color nuances and/or spatial detail to the pixelCNN.

The VAE reconstructions in Figure 5 (a) obtained using coarser color quantization carry less detail than reconstructions based on the original images using 256 color values, as expected. To understand the relatively small impact of the quantization level on the reconstruction, recall that the VAE decoder outputs the continuous means of the logistic distributions regardless of the quantization level. Only the reconstruction loss is impacted by the quantization level via the computation of the probabilities over the discrete color levels in Eq. (8). In Figure 5 (b) we observe small but consistent

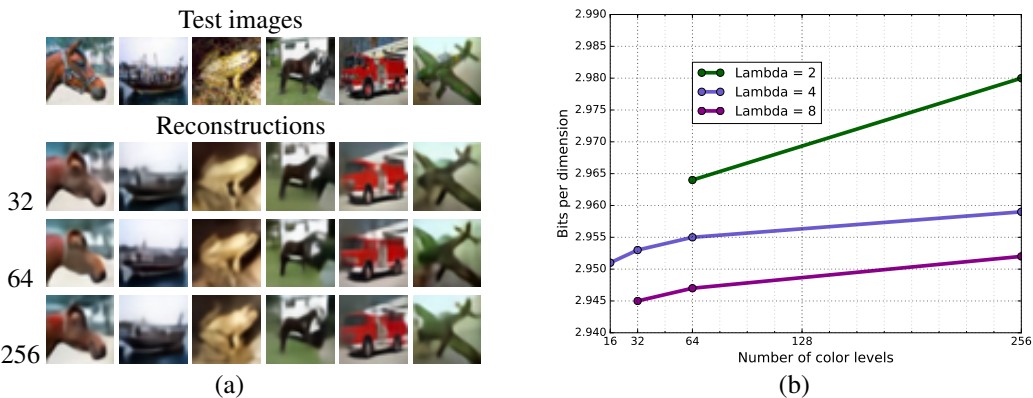

(a)                                    (b)

Figure 5: Impact of the color quantization in the auxiliary image. (a) Reconstructions of the VAE decoder for different quantization levels ($\lambda = 8$). (b) BPD as a function of the quantization level.

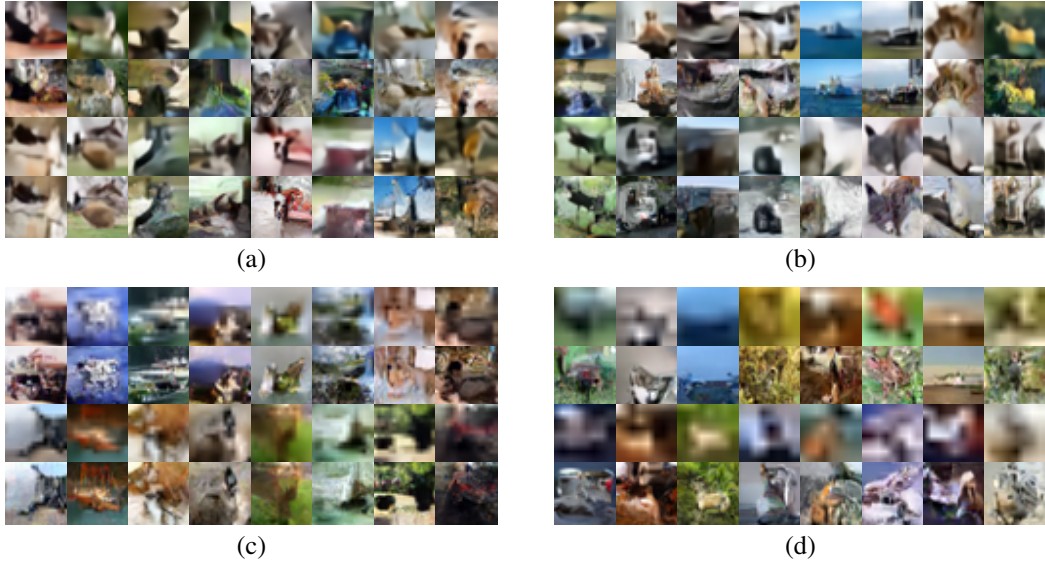

(a)                                    (b)

(c)                                    (d)

Figure 6: Samples from models trained with 32×32 auxiliary images with 256 (a) and 32 (b) color levels, and at reduced resolutions of 16×16 (c) and 8×8 pixels (d) with 256 color levels. For each model the VAE sample is displayed above the corresponding conditional pixelCNN sample.

gains in the BPD metric as the number of color bins is reduced, showing that it is more effective to model color nuances using the pixelCNN, rather than the latent variables. We trained models with auxiliary images down-sampled to 16×16 and 8×8 pixels, which yield 2.94 and 2.93 BPD, respectively. Which is comparable to the 2.92 BPD obtained using our best model at scale 32×32. In Figure 6 (a) and (b) we show samples obtained using models trained with 256 and 32 color levels in the auxiliary image, and in Figure 6 (c) and (d) with auxiliary images of size 16×16 and 8×8. The samples are qualitatively comparable, showing that in all cases the pixelCNN is able to compensate the less detailed outputs of the VAE decoder.

## 5   CONCLUSION

We presented a new approach to training generative image models that combine a latent variable structure with an autoregressive model component. Unlike prior approaches, it does not require careful architecture design to trade-off how much is modeled by latent variables and the autoregressive decoder. Instead, this trade-off can be controlled using a regularization parameter and choice

of auxiliary target images. We obtain quantitative performance on par with the state of the art on CIFAR10, and samples from our model exhibit globally coherent structure as well as fine details.

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

## A ON THE INFORMATION PREFERENCE PROPERTY

In this appendix we begin by presenting the information preference property from a bits-back coding perspective. We then show that in our setting, unlike in the standard one, it is always optimal for the autoregressive decoder to use the latent variables it is conditioned on.

Combining a VAE with a flexible decoder (for instance an autoregressive one) leads to the latent code being ignored. That is commonly attributed to optimization challenges: at the start of training $q(\mathbf{z}|\mathbf{x})$ carries little information about $\mathbf{x}$, the KL term pushes the model to set it to the prior to avoid any penalty, and training never recovers from falling in that local minimum. Chen et al. (2017) have proposed extensive explanations showing that if a sufficiently expressive decoder is used, ignoring the latents actually is the optimal behavior. Because of that, it is necessary to restrict the expressivity of the decoder so that there can be benefits from using the latents. Our main contribution is to propose a method with which it is always optimal for the autoregressive decoder to use the latents it is conditioned on regardless of the expressivity of the decoder. For self-containedness, we begin by presenting the information preference property, using the bits-back coding perspective given in Chen et al. (2017). We then show that with our formulation, using the latent variables is always the optimal behavior. This is true no matter how expressive the decoder might be, and so our solution can use autoregressive models to the full extent of their potential.

### A.1 THE INFORMATION PREFERENCE PROPERTY EXPLAINED WITH BITS-BACK CODING

Given a random variable $X$ and a a density model $p(X)$ a message $\mathbf{x}$ can be encoded with an average code length of $-\log p(\mathbf{x})$. Given an encoder $q(\mathbf{z}|\mathbf{x})$, a decoder $p(\mathbf{x}|\mathbf{z})$ and a prior $p(\mathbf{z})$, the following lossless, two-part coding scheme can be used: i) Sample $\mathbf{z} \sim q(\mathbf{z}|\mathbf{x})$ and encode it in a lossless manner [2] using $p(\mathbf{z})$. ii) Encode $\mathbf{x}$ losslessly using $p(\mathbf{x}|\mathbf{z})$. iii) The receiver can decode the message provided he has access to $p(\mathbf{z})$ and $p(\mathbf{x}|\mathbf{z})$. The expected length of the message is

$$C_{Naive} = \mathbb{E}_{\mathbf{x} \sim D, \mathbf{z} \sim q(.|\mathbf{x})}[-\log(p(\mathbf{z})) - \log(p(\mathbf{x}|\mathbf{z}))].$$

This coding scheme is inefficient because the message is encoded using $p(\mathbf{z})$ while it is sampled from $q(\mathbf{z}|\mathbf{x})$ [3]. It is necessary because the receiver does not know $\mathbf{x}$, so does not have access to $q(\mathbf{z}|\mathbf{x})$. However once the receiver has decoded $\mathbf{x}$, $q(\mathbf{z}|\mathbf{x})$ becomes available and a secondary message can be decoded from it. This yields and average code length of:

$$C_{BitsBack} = \mathbb{E}_{\mathbf{x} \sim D, \mathbf{z} \sim q(.|\mathbf{x})}[\log(q(\mathbf{z}|\mathbf{x})) - \log(p(\mathbf{z})) - \log(p(\mathbf{x}|\mathbf{z}))].$$

Notice that $C_{BitsBack}$ corresponds to the standard VAE objective. Let us examine the efficiency of this coding scheme. The lower-bound on the expected code length for the data being encoded is

---

[2]Note that $\mathbf{z}$ can be encoded without loss using any model $p(\mathbf{z})$ but the average code length $-\log(p(\mathbf{z}))$ is minimized if $p(\mathbf{z})$ matches the frequency of messages $q(\mathbf{z}|\mathbf{x})$.

[3]The coding could be optimal in the degenerate case where $p(\mathbf{z}) = q(\mathbf{z}|\mathbf{x})$, but then $\mathbf{z}$ is independent from $\mathbf{x}$ ie no information about $\mathbf{x}$ is encoded

given by the Shannon entropy: $\mathcal{H}(D) = \mathbb{E}_{\mathbf{x} \sim D}[-\log p_D(\mathbf{x})]$. The following derivation shows that the bits-back coding scheme cannot reach that lower-bound if $q(\mathbf{z}|\mathbf{x})$ is imperfect:

$$
\begin{aligned}
C_{BitsBack} &= \mathbb{E}_{\mathbf{x} \sim D, \mathbf{z} \sim q(.|\mathbf{x})}[\log(q(\mathbf{z}|\mathbf{x})) - \log(p(\mathbf{z})) - \log(p(\mathbf{x}|\mathbf{z}))] \\
&= \mathbb{E}_{\mathbf{x} \sim D}[-\log(p(\mathbf{x})) + D_{KL}(q(\mathbf{z}|\mathbf{x})||p(\mathbf{z}|\mathbf{x}))] \\
&\geq \mathbb{E}_{\mathbf{x} \sim D}[-\log(p_D(\mathbf{x})) + D_{KL}(q(\mathbf{z}|\mathbf{x})||p(\mathbf{z}|\mathbf{x}))] \\
&= \mathcal{H}(D) + \mathbb{E}_{\mathbf{x} \sim D}[D_{KL}(q(\mathbf{z}|\mathbf{x})||p(\mathbf{z}|\mathbf{x}))].
\end{aligned}
$$

If $p(.|\mathbf{x}_{j<i})$ is expressive enough, or if $q(.|\mathbf{x})$ is poor enough, the following inequality can be verified:

$$
\mathcal{H}(D) \leq \mathbb{E}_{\mathbf{x} \sim D}[-\log p(\mathbf{x}|\mathbf{x}_{j<i})] < \mathcal{H}(D) + \mathbb{E}_{\mathbf{x} \sim D}[D_{KL}(q(\mathbf{z}|\mathbf{x})||p(\mathbf{z}|\mathbf{x}))]
$$

In that case, any use of the latents that $p$ make, would make the KL divergence worse. The optimal behavior is to set $q(\mathbf{z}|\mathbf{x}) = p(\mathbf{z})$ to avoid the extra KL cost. Then $\mathbf{z}$ becomes independent from $\mathbf{x}$ and no information about $\mathbf{x}$ is encoded in $\mathbf{z}$. Let us denote $p^*(\mathbf{x}|\mathbf{x}_{j<i})$ the best autoregressive model that could be trained without using the latents. If the approximate posterior $q(.|\mathbf{x})$ is precise enough, or if $p^*(\mathbf{x}|\mathbf{x}_{j<i})$ is poor enough, then the following inequation can be verified:

$$
\mathcal{H}(D) \leq \mathcal{H}(D) + \mathbb{E}_{\mathbf{x} \sim D}[D_{KL}(q(\mathbf{z}|\mathbf{x})||p(\mathbf{z}|\mathbf{x}))] \leq \mathbb{E}_{\mathbf{x} \sim D}[-\log p^*(\mathbf{x}|\mathbf{x}_{j<i})]
$$

In that case it is possible to achieve better performance by using the latent variables. This situation can be achieved by improving $q$ or by restraining the capacity of the autoregressive model $p^*$. The conclusion is that given an encoder, the latent variables will only be used if the capacity of the autoregressive decoder is sufficiently restricted. This is the approach taken by Chen et al. (2017) and Gulrajani et al. (2017). This approach works: it has obtained competitive quantitative and qualitative performance. However, it is not satisfactory in the sense that autoregressive models cannot be used to the full extent of their potential, while learning a meaningful latent variable representation.

## A.2  OUR MODEL CIRCUMVENTS THE INFORMATION PREFERENCE PROPERTY.

We now show that in our setup, it is optimal to use the latent representations no matter how expressive the autoregressive model may be. This justifies that in theory our model should always learn meaningful latent structure.

Staying with the message coding point of view, with an AGAVE model both $(Y, X)$ have to be sent to and decoded by the receiver. If one were to only send the auxiliary representation, the setup would be that of a standard VAE: $p(\mathbf{y}|\mathbf{z})$ is factorial, and dependencies among the variables can only be modeled through the latent variables. If $\mathbf{z}$ were ignored, this would be detrimental to the reconstruction performance, and in practice VAEs with factorial decoder never ignore their latent code. Let us denote $C_{VAE}$ the expected code length required to send the auxiliary message:

$$
C_{VAE} = \mathbb{E}_{\mathbf{x} \sim D, \mathbf{z} \sim q(.|\mathbf{x})}[\log(q(\mathbf{z}|\mathbf{x})) - \log(p(\mathbf{z})) - \log(p(\mathbf{y}|\mathbf{z}))]
$$

We now show that using the latents is optimal no matter the choice of autoregressive decoder in our setup. Once $\mathbf{y}$ has been sent, sending $\mathbf{x}$ costs $\mathbb{E}_{\mathbf{z} \sim q(\mathbf{z}|\mathbf{x})}[-\sum_i \log(p(x_i|\mathbf{z}, \mathbf{x}_{j<i}))]$, so the full cost is:

$$
C_{AGAVE} = C_{VAE} + \mathbb{E}_{\mathbf{z} \sim q(\mathbf{z}|\mathbf{x})}[-\sum_i \log(p(x_i|\mathbf{z}, \mathbf{x}_{j<i}))].
$$

Suppose that the autoregressive decoder ignores $Z$. Using the fact that the Shannon entropy is the optimal expected code length, we obtain the following lower-bound, denoted $L_{\mathbf{x}}$:

$$
\begin{aligned}
C_{AGAVE} &\geq C_{VAE} + \mathcal{H}(X) \\
&= L_{\mathbf{x}}
\end{aligned}
$$

If the autoregressive decoder is allowed to use $\mathbf{z}$ a new lower-bound $L_{\mathbf{x},\mathbf{z}}$, on $C_{AGAVE}$ is obtained (attained if our decoder and coding schemes are perfect):

$$
\begin{aligned}
C_{AGAVE} &\geq C_{VAE} + \mathcal{H}(X|Z) \\
&= L_{\mathbf{x},\mathbf{z}}
\end{aligned}
$$

The entropy of a random variable decreases when it is conditioned on an other, so: $\mathcal{H}(X|Z) \leq \mathcal{H}(X)$ and $L_{\mathbf{x},\mathbf{z}} \leq L_{\mathbf{x}}$

**Conclusion:** in our setup it is always better for the autoregressive model to make use of the latent and auxiliary representation it is conditioned on. That is true no matter how expressive the model is, which is why our setup circumvents the information preference property problem. It also shows that in theory our model should learn meaningful latent structure.

## B ADDITIONAL VISUALIZATION OF PIXELCNN DECODER SAMPLES

In our model, the VAE decoder is in charge of controlling the global structure of the samples, and the autoregressive decoder is in charge of modeling low level detail. Figure 7 displays auxiliary reconstructions $f(\mathbf{z})$ and ten different samples from the autoregressive decoder conditioned on $f(\mathbf{z})$. This qualitatively shows that the low level detail added by the pixelCNN, which is crucial for log-likelihood performance, always respects the global structure of the image it is conditioned on. Moreover, the differences between the pixelCNN decoder samples also shows that the decoder is able to model variations well beyond independent per-pixel variations captured by simple factored decoders.

## C CONTROL EXPERIMENT: FINE-TUNING WITHOUT AUXILIARY LOSS

In this section we present the results of a control experiment where during training we first optimize our objective function in Eq. (7), *i.e.* including the auxiliary reconstruction term, and then switch to optimize the standard objective function of Eq. (6) without the auxiliary term.

This strong initialization could point the model towards good use of the latent variables, and circumvent optimization issues that would lead to discarding the latents. In that setting, the full model is trained to convergence then the auxiliary loss is removed and the model is fine-tuned from there. When doing this, however, the approximate posterior immediately collapses to the prior and the pixel CNN samples become independent of the latent variables.

Figure 8 displays ground-truth images, with corresponding auxiliary reconstructions and conditional samples. The reconstructions have become meaningless and independent from the ground truth images. Figure 9 displays the same behavior with samples: for each auxiliary representation four samples from the autoregressive component are displayed: they are independent from one another.

Quantitatively, the KL cost immediately drops to zero when removing the auxiliary loss, in approximately two thousand steps of gradient descent. This shows that the auxiliary loss is indeed necessary to enable the use of the latent variables by the expressive prior. Confirming the analysis of Chen et al. (2017), which we recall in Appendix A.

$f(\mathbf{z})$ Conditional PixelCNN samples

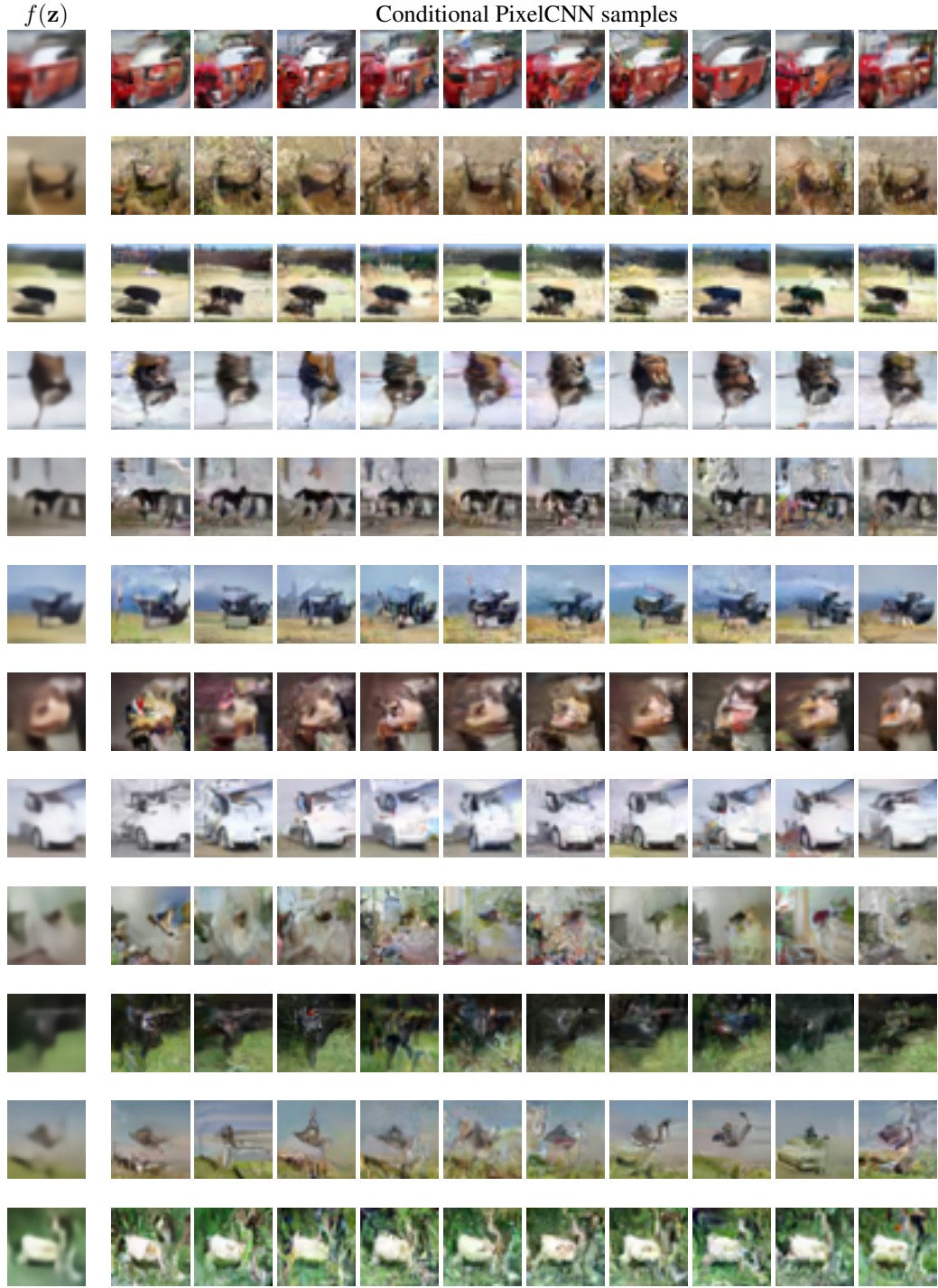

Figure 7: The column labeled $f(\mathbf{z})$ displays auxiliary representations, with $\mathbf{z}$ sampled from the unit Gaussian prior $p(\mathbf{z})$, accompanied by ten samples of the conditional pixelCNN.

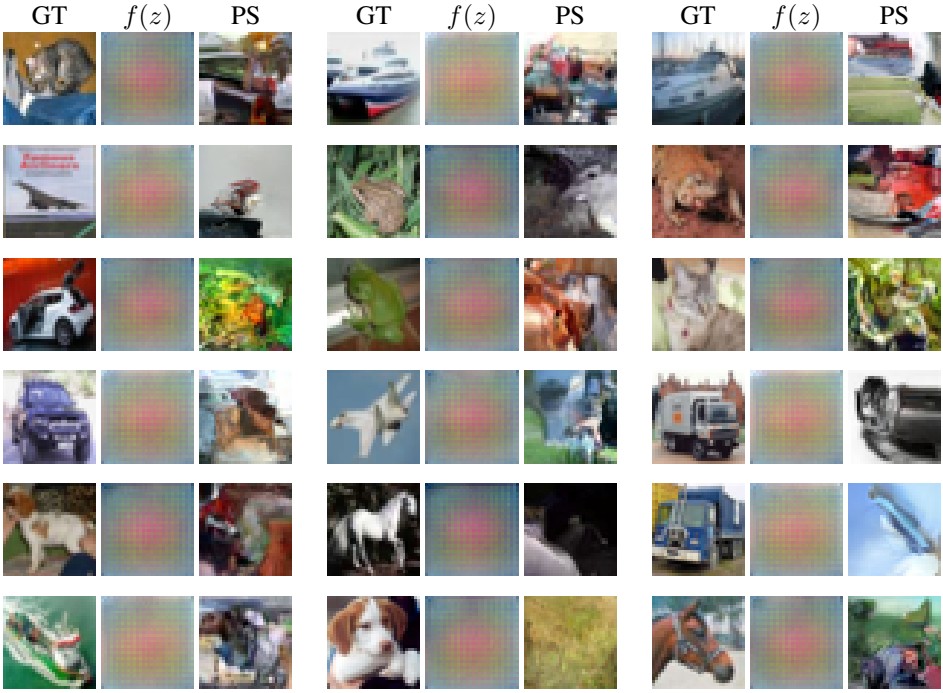

Figure 8: Auxiliary reconstructions obtained after dropping the auxilliary loss. (GT) denotes ground truth images unseen during training, $f(z)$ is the corresponding intermediate reconstruction, (PS) denotes pixelCNN samples, conditionned on $f(z)$.

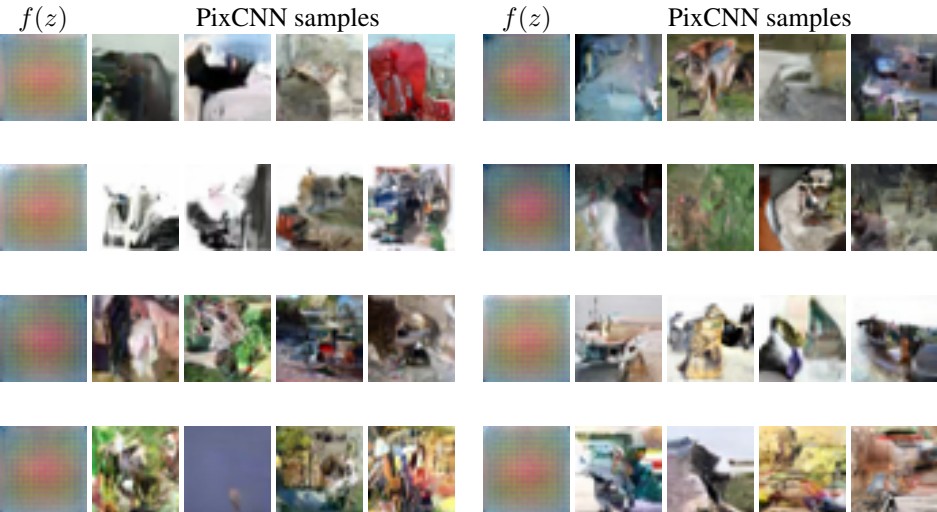

Figure 9: $f(z)$ denotes auxiliary representations obtained after dropping the auxiliary loss, where $z$ is sampled from the prior $p(z)$, and 4 corresponding samples from the pixelCNN component conditioned on $f(z)$.

