# OpenReview forum: "Auxiliary Guided Autoregressive Variational Autoencoders"
_ICLR.cc/2018/Conference — Reject_

### Official Review · AnonReviewer2 · 2017-11-26
**Important problem but no evidence of progress**

**Rating:** 5
**Confidence:** 4

**Review:**

Summary:

This paper attempts to solve the problem of meaningfully combining variational autoencoders (VAEs) and PixelCNNs. It proposes to do this by simultaneously optimizing a VAE with PixelCNN++ decoder, and a VAE with factorial decoder. The model is evaluated in terms of log-likelihood (with no improvement over a PixelCNN++) and the visual appearance of samples and reconstructions.

Review:

Combining density networks (like VAEs) and autoregressive models is an unsolved problem and potentially very useful. To me, the most interesting bit of information in this paper was the realization that you can weight the reconstruction and KL terms of a VAE and interpret it as variational inference in a generative model with multiple copies of pixels (below Equation 7). Unfortunately the authors were unable to make any good use of this insight, and I will explain below why I don’t see any evidence of an improved generative model in this paper.

As the paper is written now, it is not clear what the goal of the authors is. Is it density estimation? Then the addition of the VAE had no measurable effect on the PixelCNN++’s performance, i.e., it seems like a bad idea due to the added complexity and loss of tractability. Is it representation learning? Then the paper is missing experiments to support the idea that the learned representations are in any way an improvement. Is it image synthesis (not a real application by itself), then the paper should have demonstrated the usefulness of the model on a real task and probably involve human subjects in a quantitative evaluation.

Much of the authors’ analysis is based on a qualitative evaluation of samples. However, samples can be very misleading. A lookup table storing the training data generates samples containing objects and perfect details, but obviously has not learned anything about either objects or the low-level statistics of natural images.

In contrast to the authors, I fail to see a meaningful difference between the groups of samples in Figure 1.

The VAE samples in Figure 3b) look quite smooth. Was independent Gaussian noise added to the VAE samples or are those (as is sometimes done) sampled means? If the former, what was sigma and how was it chosen?

On page 7, the authors conclude that “the pixelCNN clearly takes into account the output of the VAE decoder” based on the samples. Being a mixture model, a PixelCNN++ could easily represent the following mixture:

p(x | z) = 0.01 \prod_i p(x_i | x_{<i}) + 0.99 \prod_i p(x_i | z)

The first term is just like a regular PixelCNN++, ignoring the latent variables. The second term is just like a variational autoencoder with factorial decoder. The samples in this case would be dominated by the VAE, which depends on the latent state. The log-likelihood would be dominated by the first term and would be minimally effected (see Theis et al., 2016). Note that I am not saying that this is exactly what the model has learned. I am merely providing a possible counter example to the notion that the PixelCNN++ has learned to use of the latent representation in a meaningful way.

What happens if the KL term is simply downweighted but the factorial decoder is not included? This seems like it would be a useful control to include.

The paper is well written and clear.

---

> ### Author Response · Authors · 2017-12-06
> **About our goal and why we believe it is achieved**
>
> Thank you for your detailed remarks and analysis of our work. We now answer your main concerns, if you want more information do not hesitate to ask.
>
> AnnonReviewer2: "To me, the most interesting bit of information in this paper was the realization that you can weight the reconstruction and KL terms of a VAE and interpret it as variational inference in a generative model with multiple copies of pixels (below Equation 7). Unfortunately the authors were unable to make any good use of this insight [...]"
>
> This insight is indeed one of the cornerstones of our contribution. Combining a factorial VAE model over the pixels with an conditional autoregressive model over another copy of the pixels naturally leads to two setting of lambda (see eq. 7 and paragraph just below), and shows that larger values of lambda also lead to valid lower bounds on the combined loss. Based on this observation, we explore models trained with different values of lambda, which improves performance from 3.2 bpd (lambda=1) to 2.92 bpd (lambda=12). The latter sets a new state-of-the-art bpd level among generative models with a non-degenerate latent variable structure. We also provide quantitative and qualitative analysis of the effect different choices of balance have on the model. In that sense, we believe the insight has been put to good use.
>
>
> AnnonReviewer2: “As the paper is written now, it is not clear what the goal of the authors is. Is it density estimation? Then the addition of the VAE had no measurable effect on the PixelCNN++’s performance, i.e., it seems like a bad idea due to the added complexity and loss of tractability. Is it representation learning? Then the paper is missing experiments to support the idea that the learned representations are in any way an improvement. Is it image synthesis (not a real application by itself), then the paper should have demonstrated the usefulness of the model on a real task and probably involve human subjects in a quantitative evaluation.”
>
> Thanks for this valuable input, that will help us to clarify the message of our paper. Our goal is to learn generative models (i.e. density estimation) with latent variable models (i.e. representation learning), for the reason stated above in response to AnnonReviewer3. Our quantitative experimental results in terms of likelihood on held-out data (the bpd metric) improve over earlier latent variable models in the literature. The images sampled from our model are used as a secondary qualitative form of evaluation. The examples in Figure 3 show that our model indeed learns a meaningful latent variable structure that is conditioning the autoregressive decoder (see also below).
>
> AnnonReviewer2: “The VAE samples in Figure 3b) look quite smooth. Was independent Gaussian noise added to the VAE samples or are those (as is sometimes done) sampled means? If the former, what was sigma and how was it chosen?”
>
> The images produced by the VAE and shown in Figure 3b) are indeed the means of the output distribution, which is why no 'salt and pepper noise' can be seen. We will clarify the text in this respect. The variance is a learned constant per color channel used across all spatial positions, and independent of the latent variable.
>
> AnnonReviewer2: “On page 7, the authors conclude that “the pixelCNN clearly takes into account the output of the VAE decoder” based on the samples. Being a mixture model, a PixelCNN++ could easily represent the following mixture: p(x | z) = 0.01 \prod_i p(x_i | x_{“
>
> The end of this argument was unfortunately cut off. Can you please send an update with the complete text?
>
> In the meantime, let us respond as follows. The samples in figures 1c, 3 and 6 show the intermediate representation f(z) that is computed by the VAE decoder, together with samples from the pixelCNN decoder that is conditioned on f(z), see also Figure 2 for schematic overview.
> Let us suppose, contrary to our claim, that the pixelCNN decoder ignores the output of the VAE decoder, i.e. that the pixelCNN output is independent of the VAE output. In this case the image pairs of VAE decoder output and PixelCNN sample should not correlate at all in figures 1c, 3 and 6. Yet, we observe in each single example a clear correspondence between the pixelCNN sample and the conditioning VAE output. The latter looks like a smoothed version of the former. Therefore, we conclude that the latter is not independent of the former, and that pixelCNN does take into account the VAE output, i.e. we succeed in conditioning the pixelCNN on the VAE output.
> We hope this clarifies our statement, and we will update the text accordingly.

---

> ### Author Response · Authors · 2017-12-15
> **On why it is optimal to use the latent variables in our setting**
>
> Thank you for your update, we now answer the additional points raised.
>
> Let us briefly recall the argument of [Theis et al., 2016] on possible mismatches between sample quality and log-likelihood scores of models. Theis et al, present a hypothetical mixture model p(x) = a p1(x) + (1-a) p2(x), where p1 attains a good likelihood but samples of poor visual quality, and p2 gives a poor likelihood but samples of good visual quality. With small a, e.g. a=0.001, almost all samples will look good. Yet the model can still attain a good likelihood since log(p(x)) >= log(a) + log(p1(x)), and for high dimensional x the log(a) term will be negligible compared to the log-likelihood term log(p1(x)).
> Applied to our setting, it could be that we obtain good likelihood scores with such a mixture, in which p1 would be a pixelCNN ignoring the latent variable, and p2 would generate samples that correlate with our intermediate latent variable representation f(z) (see Fig 2 in the paper).
>
> Let us first make a general remark about the mixture example of Theis et al. Although such a hypothetical mixture can in principle obtain a “good” likelihood score (relatively close to that of p1), there is an incentive in the training objective not to converge to such solutions. First, the second component p2 is assumed to have poorer log-likelihood than p1. The mixture is therefore a suboptimal solution, in particular a better solution would be obtained by just using the first mixture component p1. Second, assume that we do have a solution that includes the mixture component with good samples, despite being detrimental to the likelihood. Given that it is non-trivial to obtain a model that generates samples with good visual quality, finding such a solution would require another training signal, which we do not have in our model.
>
> In addition, we now explain why in our approach, no matter how expressive the autoregressive decoder is,  to minimize the loss it is optimal to use the latent variables. Therefore, in the hypothetical mixture of Theis et al, p1 should generate samples that correlate with the latent variable, unless it is also suboptimal in terms of the loss.
>
> For the standard VAE loss, it has been shown that given a sufficiently expressive decoder, it is optimal to ignore the latent variables. See for example [Variational Lossy Autoencoder, Chen et al] for a theoretical justification based on bits-back coding. The main argument is that, under the evidence lower bound objective function of eq (2) in our paper, for a model without latent variables the KL penalty can be trivially set to zero since in that case p(z|x) = p(z) and we can choose q(z)=p(z), while this is not the case for a model that does exploit latent variables for which p(z|x) can be arbitrarily complex depending on the relation between x and z. Therefore, models without latent variables are favored, as long as they obtain a log-likelihood that is the same as a model with latent variables, or worse up the the KL cost of the latent variable model.
>
> Our setting is different: the intermediate auxiliary loss ensures a meaningful latent variable since the factored decoder can only model variable dependencies through the latent variables. This is what underlies any non-degenerate variational autoencoder model with a decoder of limited capacity. Given that the factored decoder induces a certain non-zero KL “cost” in eq (2), there is no longer an advantage in ignoring the latent variable for the autoregressive decoder. Indeed, since the factored decoder renders x and z dependent, the uncertainty on x is reduced by conditioning on z. Therefore, it is optimal for the autoregressive decoder to exploit the information on x carried by the posterior on the latent variable q(z|x),
>
> We will clarify these points in the paper.
>
> “ What happens if the KL term is simply down-weighted but the factorial decoder is not included? This seems like it would be a useful control to include.”
>
> Simply down-weighting the KL term (to less than 1) would indeed encourage the decoder to use the latent variables. The reconstruction quality would clearly improve, however in that case the loss would no longer be a lower-bound on the log likelihood of the model. We have performed a control similar to what you suggest: we trained our model to convergence, then removed the auxiliary loss and tried fine-tuning from there. This strong initialization could point the model towards good use of the latent variables. When doing this, however, the encoder posterior immediately collapses to the prior and the pixel CNN samples become independent of the latent variables. This shows that the auxiliary loss is necessary to enable the use of the latent variables by the expressive prior. We will include visualizations of this control in a revised version of the paper shortly.

---

> ### Comment · AnonReviewer2 · 2018-01-12
> **Responding to authors**
>
> My main problem is still that it's not clear what this model has to offer. The model is neither able to improve density estimation over PixelCNNs (while adding complexity), nor has it been shown to learn better representations (none of the evaluations seem appropriate to evaluate representations). Nevertheless, I slightly revised my score.
>
> > First, the second component p2 is assumed to have poorer log-likelihood than p1. The mixture is therefore a suboptimal solution, in particular a better solution would be obtained by just using the first mixture component p1.
>
> This reasoning is incorrect. The log-likelihood of p1 + p2 may be minimally better than p1's, even if p1 has a much better log-likelihood than p2. Note that the log-density of the mixture is approximately the maximum of the individual log-densities (see log-sum-exp as approximation to maximum), so using p2 generally comes at little cost to the model. Log-likelihood does not even create a big incentive to have a small weight on p2.
>
> > Our quantitative experimental results in terms of likelihood on held-out data (the bpd metric) improve over earlier latent variable models in the literature.
>
> Density estimation using latent variable models is not a well defined goal – a PixelCNN and in fact any autoregressive model can be equivalently formulated/viewed as a latent variable model (and if it weren't, we can always mix in a latent variable model at near-zero cost to the log-likelihood) – nor is it clear why it is a desirable goal.

---

### Official Review · AnonReviewer1 · 2017-11-27
**Good paper**

**Rating:** 7
**Confidence:** 4

**Review:**

The proposed approach is straight forward, experimental results are good, but don’t really push the state of the art. But the empirical analysis (e.g. decomposition of different cost terms) is detailed and very interesting.

---

> ### Author Response · Authors · 2017-12-06
> **Answer to the review**
>
> Thank you for your appreciation of our analysis and empirical evaluation.
>
> AnnonReviewer1: “The proposed approach is straight forward, experimental results are good, but don’t really push the state of the art. But the empirical analysis (e.g. decomposition of different cost terms) is detailed and very interesting.”
>
>  We provide justifications of why we believe that our work significantly pushes the state of the art in latent variable density modeling in our other answers, and hope these arguments are satisfying.

---

> ### Comment · AnonReviewer1 · 2018-01-11
> **Updated paper**
>
> Hello Everyone,
>
> I read the updated version of the paper and went through the discussion here in the comment section. As mentioned before: the paper does not significantly push the state of the art for density modelling and the empirical results do not outperform pure autoregressive approaches.
>
> Nevertheless, I think there is a active community of researches interested in combining latent variable models with autoregressive decoders for various reasons (e.g. sampling runtime performance; using latent representations for related tasks; etc. ). I agree with Reviewer2 that this paper does not solve this issue, but I think it contributes to
> the ongoing discussion in the field. And compared to [Chen et al. '17, Kolesnikov & Lampert '17, Reed et. al. 17, etc.] it does provide a new perspective: the interpretation of a generative model for replicated pixels.
>
> I think this perspective deserves to be heard and I will therefore maintain my rating (good paper, accept)

---

### Official Review · AnonReviewer3 · 2017-11-29
**AUXILIARY GUIDED AUTOREGRESSIVE VARIATIONAL AUTOENCODERS**

**Rating:** 5
**Confidence:** 4

**Review:**

The authors present Auxiliary Guided Autoregressive Variational autoEncoders (AGAVE), a hybrid approach that combines the strengths of variational autoencoders (global statistics) and autorregressive models (local statistics) for improved image modeling. This is done by controlling the capacity of the autorregressive component within an auxiliary loss function.

The proposed approach is a straightforward combination of VAE and PixelCNN that although empirically better than PixelCNN, and presumably VAE, does not outperform PixelCNN++. Provided that the authors use PixelCNN++ in their approach, quantitively speaking, it is difficult to defend the value of adding a VAE component to the model. The authors do not describe how \lambda was selected, which is critical for performance, provided the results in Figure 4. That being said, the contribution from the VAE is likely to be negligible given the performance of PixelCNN++ alone.

- The KL divergence in (3) does more than simply preventing the approximation q() from becoming a point mass distribution.

---

> ### Author Response · Authors · 2017-12-06
> **On the value of adding the VAE component, and the selection of lambda**
>
> Thank you for your constructive review of our work. We will address the main concerns raised, please do not hesitate to ask for more detail if needed.
>
> AnnonReviewer3: “ [...] empirically better than PixelCNN, and presumably VAE, does not outperform PixelCNN++.”  “ Provided that the authors use PixelCNN++ [...] it is difficult to defend the value of adding a VAE component to the model”
>
> Our main contribution is a method based on an auxiliary loss to learn generative models that combine a non-degenerate latent variable structure with expressive autoregressive decoders.
>
> Among latent variable models, our model sets a new state-of-the-art result of 2.92 bpd. That is  the same score as obtained by pixelCNN++ which does not learn a latent variable representation. Among VAE models with factored observation model of p(x|z), VAE-IAF obtains the best quantitative score of 3.11 bpd on CIFAR10. Our performance of 2.92 bpd represents an important improvement. The improvement over Lossy-VAE (2.95 bpd), former best model with latent variables, is 0.03 bpd. Our auxiliary loss allows us to use a more powerful autoregressive decoder, which allows us to improve over the Lossy-VAE result. The numbers that support these claims can be found in Table 1. We will improve the presentation of Table 1 to highlight which models use latent variables and/or autoregressive decoders, to more easily appreciate our contribution in terms of the quantitative evaluation results.
>
> Unlike autoregressive models such as pixelCNN++, latent variable models learn data representations which are useful for tasks such as semi-supervised learning, see e.g. (Kingma et al., NIPS 2014). Therefore we believe that our work makes an important contribution to generative representation learning.
>
> AnnonReviewer3:  “The authors do not describe how lambda was selected, which is critical for performance [...]”
>
> The results used when comparing with the state of the art are reported using our best configuration with lambda equal to 12. We will clarify this in the text. In figures 3, 4 and 5 qualitative and quantitative results are reported, as indicated, over a range of lambda values. The choice of lambda is important, but not critical, provided it is taken to be bigger than two: Figure 4 shows that beyond the first drop of 0.20 bpd  when going from lambda =1 to lambda = 2, the bpd further decreases monotonically to down to an improvement of 0.26 bpd  for lambda=12.

---

### Public Comment · ~Lucas_Caccia1 · 2017-11-29
**Implementation details**

Hi, I'm a Master's student in Computer Science taking part in the ICLR reproducibility challenge, and I have a few questions regarding implementation.

First, you mention that "We allow each layer of the pixel-CNN to take additional input using non-masked convolutions from the feature stream based on the VAE output". How is this implemented exactly ? Looking at the PixelCNN++ implementation, we see that every layer is composed of 2 streams of (n=5) resnet blocks, specifically one downwards and one downward+rightward stream. In AGAVE, do you feed the vae output to every resnet block in both streams ? or do you you feed it to the top resnet block at every layer for both streams ?

Second, you mention that you condition on  the "VAE decoder output f(z), or possibly an upsampled (downsampled?) version if y has a lower resolution than x".  How exactly do you perform downsampling ? do you use strided convolutions ? If so,  do you use the same downsampled vae output for the kth and the (K-k)th layer of the PixelCNN, as they have the same resolution?

Third, did you use the default hyperparameters proposed on the repositories of IAF/VAE and PixelCNN++ ? If not, what modifications did you make? Did you reduce the size of the networks so that they both fit on a single GPU ? What kind of initialisation was performed on the weights ?

Lastly, for how long were the PixelCNN and IAF/VAE models pretrained ? Do you have any other advice/specifications for people aiming to reproduce your results ?


Many thanks

---

> ### Author Response · Authors · 2017-12-08
> **Precisions about implementation details**
>
> Hello,
>
> Your questions are very relevant, and will be helpful to improve reproducibility, thank you for your interest. We will release the code if the paper is accepted. In the meantime, we'll do our best to answer your questions. Do not hesitate to ask if you need more information.
>
> "We allow each layer of the pixel-CNN to take additional input using non-masked convolutions from the feature stream based on the VAE output". How is this implemented exactly ? Looking at the PixelCNN++ implementation, we see that every layer is composed of 2 streams of (n=5) resnet blocks, specifically one downwards and one downward+rightward stream. In AGAVE, do you feed the vae output to every resnet block in both streams ? or do you you feed it to the top resnet block at every layer for both streams ?"
>
> In Agave we use separate stream for the conditional information: the coarse image is given as input to that new stream at the first layer, then this new stream is used as input to the stream looking left. This is different from other methods of conditioning used with pixelCNN and pixelCNN++, we find it quite natural.
> The new stream does not take other inputs, so convolutions do not have to be masked. You could chose to also give other streams as input to the new one, provided that the convolutions across the new inputs are masked, and that if the stream looking to the left is used as input to the conditioning, then the conditioning is not used as input to the stream looking up (that would 'look into the future').
>
> "Second, you mention that you condition on  the "VAE decoder output f(z), or possibly an upsampled (downsampled?) version if y has a lower resolution than x".  How exactly do you perform downsampling ? do you use strided convolutions ? If so,  do you use the same downsampled vae output for the kth and the (K-k)th layer of the PixelCNN, as they have the same resolution?."
>
> The choice of downsampling method (max pooling, average pooling, strided convolution) is not critical for log-likelihood performance, the results reported all use average pooling. The second part of your question is answered in our previous answer: $y$ is given as input at the top of the stream. The new stream is downsampled at the same time as the other ones.
>
> "Third, did you use the default hyperparameters proposed on the repositories of IAF/VAE and PixelCNN++ ? If not, what modifications did you make? Did you reduce the size of the networks so that they both fit on a single GPU ? What kind of initialisation was performed on the weights ?"
>
> The hyperparameters are modified as little as possible. Changing the depth of the pixelCNN is an option that we explored. It slightly hurts performance for instance by using 3 resnet blocks instead of 5 we end up with 2.96 bpd, but does bring the size of the model down. One of our goal was to show that our method can be used without restraining the autoregressive component, though, so this was done more as a sanity check to confirm that it is indeed useful to use the full model. If you are working with the VAE at a downsampled scale, the memory cost of this component is greatly reduced. If size on the GPU is an issue (it was for us, as we trained with 1 GPU only) you have two options: pretrain each component separately without compromise, then finetune the two together with batch sizes small enough to fit your GPU, or train both components together with a reduced batch size to begin with. Weight normalisation was used in our experiments, and we used the associated initialisation.
>
> "Lastly, for how long were the PixelCNN and IAF/VAE models pretrained ? Do you have any other advice/specifications for people aiming to reproduce your results ?"
>
> Approximately 2 days for the VAE, and 3 days for the pixelCNN when pretraining sequentially. If you pretrain the pixelCNN on the downsampled GT, training time of the pixelCNN will be greatly reduced though we did not do so in our final experiments to not overly complicate the training procedure. If you are working at a reduced scale for the VAE, a day is more than enough. General advice could include: displaying a lot of curves (breakdown of the different cost functions), visualizing intermediate representations and intermediate targets as well as final ones, and having a minimal version of the architecture to debug, as pixelCNN++ is a heavy model.
>
> We hope this help, don't hesitate if you need more information or if something is unclear.

---

> > ### Public Comment · ~Lucas_Caccia1 · 2017-12-10
> > **Final questions**
> >
> > Hi,
> >
> > thanks for taking the time to answer my questions, it is much appreciated. I have a few follow up questions if you don't mind :
> >
> > From what I understand, the vae stream is essentially the same as the other 2 streams, with the difference that the convolutions are not masked. Does this stream have as many layers as the other 2 ? In other words, does this stream also have 6 layers of n resnet blocks? What kernel size was used for the non masked convolutions ?
> >
> > Second, you connect the vae stream with the rest of the network by letting the stream looking left  take input from the vae stream. Is this connection similar to how the left stream takes input from the up stream (a non-linearity followed by network-in-network (1x1 conv) connection) ?
> >
> > Thanks again

---

> > > ### Author Response · Authors · 2017-12-11
> > > **About the new stream**
> > >
> > > Hi,
> > >
> > > You are welcome, don't hesitate to ask.
> > >
> > > " From what I understand, the vae stream is essentially the same as the other 2 streams, with the difference that the convolutions are not masked. Does this stream have as many layers as the other 2 ? In other words, does this stream also have 6 layers of n resnet blocks? What kernel size was used for the non masked convolutions ? "
> > >
> > > The non masked convolutions use a kernel of size 3x3. You are correct, it is exactly the same design as the two other streams: everywhere there is a block for the two others, there is one for the new stream, and down sampling is done at the same time. The main reason for this is simplicity, and that's why we preferred it over other means of conditioning a pixelCNN that are used in other papers.
> > >
> > > "Is this connection similar to how the left stream takes input from the up stream ?"
> > > Yes, it is done exactly in the same way. Also, the conditioning stream and the one looking up are given to the one looking left by first concatenating them.
> > >
> > > Good luck for the challenge

---

### Author Response · Authors · 2017-12-06
**Thank you for the reviews**

Dear reviewers, thank you for the constructive feedback. We discuss the concerns raised in separate answers to each review, aiming for brevity and clarity in this first response. Please don’t hesitate to let us know if you feel certain points should be discussed in more detail. We will revise the pdf based on your feedback over the coming days.

Update:
The paper has been updated to take into account major remarks made by the reviewers. The appendix now contains a presentation of why it is always optimal for the auto-regressive decoder to use the latent variables no matter its expressivity, an additional visualization where the auxiliary representation is fixed and multiple images are sampled from the auto-regressive component, and a control experiment in which the auxiliary loss is removed after pre-training.

---

### Author Response · Authors · 2018-01-04
**Paper updated**

Dear reviewers,

The paper has been updated to take into account major remarks made. The appendix now contains a presentation of why it is always optimal for the auto-regressive decoder to use the latent variables no matter its expressivity, an additional visualization where the auxiliary representation is fixed and multiple images are sampled from the auto-regressive component, and a control experiment in which the auxiliary loss is removed after pre-training.

Again, thank you for your feedback and time spent reviewing this paper.

---

### Decision · Program_Chairs · 2018-01-29
**ICLR 2018 Conference Acceptance Decision**

**Decision:**

Reject

**Comment:**

To ensure that a VAE with a powerful autoregressive decoder does not ignore its latent variables, the authors propose adding an extra term to the ELBO, corresponding to a reconstruction with an auxiliary non-autoregressive decoder. This does indeed produce models that use latent variables and (with some tuning of the weight on the KL term) perform as well as the underlying autoregressive model alone. However, as the reviewers pointed out, the paper does not demonstrate the value of the resulting models. If the goal is learning meaningful latent representations, then the quality of the representations should be evaluated empirically. Currently it is not clear whether that the proposed approach would yield better representations than a VAE with a non-autoregressive decoder or a VAE with an autoregressive decoder trained using the "free bits" trick of Kingma et al. (2016). This is certainly an interesting idea, but without a proper evaluation it is impossible to judge its value.